# Metabolic syndrome and related factors in Cameroonian women under contraceptive use

**Dandji Saah Marc Bertrand**[1]*, **Dangang Bossi Donald Séverin**[1], **Tanguenan Floraise Lynda**[2], **Zambou Ngoufack François**[1,2]

1 Research Unit of Biochemistry, Medicinal Plants, Food Sciences and Nutrition (URBPMAN), Department of Biochemistry, Faculty of Science, University of Dschang, Dschang, Cameroon, 2 Department Public Health, Faculty of Medicine and Pharmaceutical Sciences, University of Dschang, Dschang, Cameroon

* dandjisaah@yahoo.fr

## Abstract

### Objectives

Contraceptive is a device or drug that prevents a woman from becoming pregnant. Some types of contraceptive can cause a myriad of secondary effects such as overweight, increase of blood pressure triglycerides, and glucose intolerance. The combination of these secondary effects could, in the long term, develop metabolic syndrome in these women. The purpose of this study was to determine the prevalence of metabolic syndrome and related factors in Cameroonians women on contraceptives.

### Methods

This was a cross-sectional study that included 231 Cameroonians fasting women from 18 to 49 years of age, on contraceptives. Sociodemographic, lifestyle, anthropometric and biochemical characteristics were collected. Metabolic syndrome was diagnosed using the criteria of the National Cholesterol Education Program- Adult Treatment Panel III. After validation of the data, statistical analysis was performed using Epi-Info software version 7.2.2.16 and the statistical level of significance was set at 5.0%.

### Results

231 were using a modern contraceptive method, 28 were not using a contraceptive method, and 12 were non-consenting. The contraceptive method use rate was 89.19% and the most commonly used method was injectable contraception (49.35%). According to National Cholesterol Education Program criteria, almost 38.96% of these women were overweight and 50.65% had a serum high density lipoproteins cholesterol level of less than 0.50 g/L. Among women on contraceptives, the prevalence of metabolic syndrome was 22.08%. However, there was no significant association between contraceptive use and the occurrence of metabolic syndrome (p = 0.63).

**Data Availability Statement:** All relevant data are within the manuscript and its Supporting Information files.

**Funding:** The author(s) received no specific funding for this work.

**Competing interests:** The authors have declared that no competing interests exist.

## Conclusion

Contraceptive use was certain in all the participants, it's reported that, according to the NCEP-ATPIII a prevalence of 22.08% of metabolic syndrome among women using modern contraceptive methods in Douala, Republic of Cameroon. The high-risk groups were women using injectable method. Therefore, lipid profiles should be assessed in those women in order to manage them better.

## Introduction

Contraception refers to the means of preventing sexual relations from leading to pregnancy. According to the World Health Organization (WHO), it is defined as the use of agents, devices, methods, or procedures that reduce or prevent the probability of conception [1]. Among the methods of contraception, we distinguish firstly the natural methods (periodic abstinence, temperature method, Billings method, fixed day method, breastfeeding method, amenorrhea and coitus interrupted). Secondly, the barrier methods (male and female condoms, cervical cap, diaphragm, spermicides). Thirdly, the hormonal methods (pills, patches, implants, injectables, hormonal Intra-Uterine Device (IUD)) and finally, non-hormonal methods (copper IUD, voluntary surgical contraceptive methods) as well as emergency contraception [2]. These methods have different modes of action and are more or less effective in preventing unwanted pregnancies [3].

According to an estimation in 2021 by Bekele et al., the proportion of women using contraception is about 62%, with 38% injectables, 22% implants and 1% pills [4]. The percentage of women on contraceptives of childbearing age (15 to 49 years) using a contraceptive is 19% in Central Africa [5]. The annual progress report of Family Planning in Cameroon in 2020 indicates that 18.7% of women on contraceptives of childbearing age use a modern method of contraception [6]. These modern methods of contraception affect different people in different ways. As well as helping to prevent unwanted pregnancies, they can cause a wide range of secondary effects. These can include overweight, increased blood pressure, carbohydrate intolerance, and increased triglycerides [7]. The combination of these secondary effects could, in the long term, develop metabolic syndrome (MetS) in these women. MetS is a common public health problem worldwide and particularly in Cameroon. Its incidence gradually increasing both in women and men in our powerless look.

The MetS is characterized by a constellation of physiological, biochemical and asymptomatic abnormalities which can coexist with genetic and inherited factors. Thus, according to the definition that we chose for our study, it consists in the same woman to the diagnosis of at least 3 of the 5 following criteria: Waist circumference greater than 88 cm, high density lipoproteins cholesterol (HDL-c) less than 50 mg/dl, fasting triglycerides equal or greater than 150 mg/dl, blood pressure equal or greater than 130/85 mmHg and fasting blood glucose equal or greater than 110 mg/dl.

Constantly increasing, the metabolic syndrome represents a social and economic disaster for our countries' healthcare systems [8]. Similarly, global data on metabolic syndrome remain difficult to obtain, but it is estimated that over a billion people worldwide are affected by metabolic syndrome [9]. Both cohorts showed a trend towards the increase of MetS components moving from the lowest to the highest CV risk class, with a high prevalence of patients with four or five MetS determinants allocated in the high/very high CV risk group [10].

Today, due to the increase of the use of contraceptives among women and their sedentary lifestyle, occurrence of metabolic syndrome in these women is a major public health challenge.

This justifies our study on metabolic syndrome in these women using contraceptive methods. The number of studies conducted to investigate any relationship between the use of contraceptives and metabolic syndrome is very limited. Therefore, this study was designed to evaluate the implication of contraceptive use on women's health especially on metabolic syndrome.

## Materials and methods

Data collection of this cross-sectional study took place among 231 women on contraceptives aged 18 to 49 years. It took place from September 6, 2021 to February 15, 2022 in four hospitals in the town of Douala and was randomly chosen. A self-administered questionnaire titled « *Women's Lifestyle* » was developed. Women were provided with an information sheet and consent form to fill before completing the self-report questionnaire. To ensure that women volunteering could take part. That is, they were informed that their participation was both voluntary and anonymous. In addition, their anthropometrics data, lifestyle factors and biochemical parameters were collected.

### 1. Study population, recruitment and eligibility

Our target population was Consenting women of childbearing age using a modern contraceptive method (pill, implant, injectable, hormonal IUD) for at least three months. Women on medication that could affect biochemical parameters were excluded from the study. Participants were recruited by the principal investigator in the various health facilities in accordance with the ethical rules governing human research in Cameroon. We had contacted 271 women, of whom 231 were on modern contraceptives, 28 were not using contraceptives, and 12 did not consent to the study. Thus, they were recruited in health centre's between 8am and 12pm, and at the interview, we informed the participants about the purpose, methodology, and constraints of the study prior to the blood draw; we made it clear to them that their blood sample would be destroyed at the end of the analysis and the volunteers signed the informed consent form. This study was approved by the ethics committee of the University of Douala and was conducted in compliance with the Declaration of Helsinki. The sample size was calculated using the following standard Lorentz formula: $[N = t^2 \times P \times (1\text{-}P) / m^2]$. Where N is the minimum sample size to obtain significant results for an event and a fixed level of risk; t is the confidence level (typical value of the confidence level of 95% is 1.96); p is the estimated prevalence of metabolic syndrome, based on the literature 38.98% mortality rate due to metabolic syndrome in Cameroon [11] and m is the margin of error (usually set at 5%). After calculation, the minimum population size was 233.

### 2. Questionnaire

Only an interviewer who was trained and standardised in administering the structured questionnaire was employed. Demographic characteristics: demographic information was collected, sex, occupation, marital status, level of study. Lifestyle related habits: including questions regarding physical activity, eating, smoking and alcohol drinking habits. The clinical and complete examination included also two measurements of blood pressure which was recorded electronically and anthropometric measurements as self-reported height and weight (from which BMI was calculated) and waist circumference (WC) were done.

### 3. Blood collection and laboratory analysis

Prior to collection, the tubes were labelled with the patient's identification information and date of collection. Five millilitres (5ml) of blood were collected in a dry tube by venepuncture

from brachial vein following standard infection prevention procedures at the elbow for the determination of HDL-c and triglycerides. A drop of blood taken from the pulp of the finger with a lancet was used for the determination of blood glucose. All samples were taken under strict aseptic conditions. The samples were transported to the laboratory at 18 to 25°C, using a dedicated cooler, within 6 hours. The blood was then centrifuged at 3000 rpm for 5 minutes and aliquoted into Eppendorf tubes (600 μl/tube) before being stored at -20°C until the assays.

## 4. Metabolic syndrome criteria

**4.1. Anthropometric parameters measurement.** The anthropometric parameters included weight, height and waist circumference were measured using standard measurement tools. Weight in kg was measured with an electronic scale to the nearest 0.1 kg. The height in meter was measured using a portable stadiometer to the nearest 0.5 cm, with subjects standing upright on a flat surface without shoes. The weight and height were used to calculate the BMI that is the ratio of weight (kg) over height in meter square and BMI categories were defined as underweight with BMI $\leq$ 18.5 kg/m$^2$; normal was BMI 18.5–24.9 kg/m$^2$; overweight was BMI 25–29.9 kg/m$^2$ and obese was BMI $\geq$ 30 kg/m$^2$. To measure the WC, the subject stands comfortably, weight evenly distributed between both feet, feet slightly apart. The measurement is taken in a horizontal plane on the smallest part of the waist, equidistant between the bottom of the ribs and the pelvis bone [12]. The recommended WC thresholds for assessing abdominal obesity for sub-Saharan African women was $\geq$88 cm [13].

**4.2. Blood pressure measurement.** Diastolic and systolic blood pressure (BP) measurements were performed during the interview using OMRON automatic BP monitor (model: M3; HEM-141-E, serial n°: 20170916247VG, Japan) after a rest period of 5–10 minutes in a sitting position. Two measurements were performed on each subject and the mean value was considered. Elevated BP as a component of MetS was defined as $\geq$ 130/85 mmHg. A systolic BP of 120–129 mmHg and/or diastolic BP of 75–79 mmHg were considered as prehypertension. Hypertension was defined as systolic BP $\geq$ 130 mmHg and/or diastolic BP $\geq$ 85 mmHg [14].

**4.3. Biochemical variables measurement.** Fasting blood glucose samples were collected using an Accu Chek® Active blood glucose meter and test strips. A small sample of blood was taken from the fingertip with a lancet and then placed on the strip for blood glucose measurement. Triglycerides (TG) and HDL-c levels were drawn and analysed following the overnight fasting of 12 hours. Each blood sample was labelled with the participant's number to avoid recording errors. Blood glucose values were defined as normal between 1 and 1.25 g/l and hyperglycaemia > 1.25 g/l. High TG was defined as TG between 0.4 and 1.4 g/l, regardless of gender, and hypertriglyceridemia > 1.4 g/l. In contrast, low HDL-c was defined as HDL-c below 0.5 g/l in women [12].

## 5. Statistical analysis

Statistical analysis was performed using the Epi info 7.2.2.16. For quantitative variables, data was presented as mean ± standard deviation (SD). The comparison between groups was done by Mann-Whitney test. All analyses were two-tailed, and « *p* » values less than 0.05 were considered statistically significant.

## 6. Ethics approval

Ethical approval to conduct this study was obtained from Institutional Ethics Committee for Research on Human Health of Douala University-Cameroon (approval number 2553CEI-Udo/06/2021/M). Consent was obtained from the study participants prior to data

collection after an explanation on study aim and objectives. The participants were assured of confidentiality, privacy, anonymity and non-coercive nature of the study.

## Results

### 1. Sociodemographic and clinical parameters related to the prevalence of MetS determinants

From the 231 women surveyed, 85.24% of childbearing age using a modern contraceptive method between the ages of 18 and 49 were included in this study. Among them, 49.35% were married, 36.36% were university graduates, 59.74% were Catholic faith and their contraceptive preferences that were injectable method (49.35%). The background characteristics of this study population are shown in Tables 1 and 2.

### 2. Metabolic syndrome variables according to NCEP-ATPIII criteria

The different variables between "in good health" and "not in good health" subjects are presented in Table 3. Out of two hundred and thirty-one women, 36.37% (95.04 ±6.38) were obese; 58.44% (85.62 ±4.20) were hypertensive (DBP) and 50.65% (0.45 ±0.05) had dyslipidaemia due to HDL-c imbalance. However, the results show that 94.80% (0.78 ±0.10) of those women have good blood sugar levels. The means (±SD) of the components of MetS including other related variables like overweight, systolic blood pressure, total cholesterol, LDL-c and triglycerides values were measured among the study population.

**Table 1.  Sociodemographic characteristics of the study population.**

| Variables | Category | n (%) |
|---|---|---|
| Age (year) | 15–24 | 24 (10.39) |
| | 25–34 | 126 (54.55) |
| | 35–44 | 69 (29.87) |
| | > 45 | 12 (5.19) |
| Marital status | Single/Divorced/Widowed | 117 (50.65) |
| | Married/Live together | 114 (49.35) |
| Occupation | Household | 63 (27.27) |
| | Civil servant | 54 (23.38) |
| | Traders | 21 (9.09) |
| | Others | 93 (40.26) |
| Education | No formal education | 6 (2.60) |
| | Primary | 24 (10.39) |
| | Secondary | 120 (51.95) |
| | University | 81 (35.06) |
| Religion | Catholic | 138 (59.74) |
| | Protestant | 66 (28.57) |
| | Muslim and others | 27 (11.69) |
| Use or non-use of modern contraception | Not used | 28 (10.81) |
| | Used | 231 (89.19) |
| Type of contraceptive used | Implant | 90 (38.96) |
| | IUD | 27 (11.69) |
| | Injectable | 114 (49.35) |
| | Pills | 0 (0) |

n = Frequency; % = Percentage; IUD = Intra-Uterine Device

**Table 2. Prevalence of MetS determinants according to selected sociodemographic and clinical parameters.**

| Variable | Category | Obese | | P-value | Diabetic | | P-value | Hypertensive | | P-value | Dyslipidemia | | P-value |
|---|---|---|---|---|---|---|---|---|---|---|---|---|---|
| | | Yes | No | | Yes | No | | Yes | No | | Yes | No | |
| | | n (%) | n (%) | | n (%) | n (%) | | n (%) | n (%) | | n (%) | n (%) | |
| Age (year) | 15–24 | 9 (37.50%) | 15 (62.50%) | 0.826 | 3 (12.50%) | 21 (87.05%) | 0.768 | 6(25%) | 18(75%) | 0.248 | 12(50%) | 12(50%) | 0.085 |
| | 25–34 | 42 (33.33%) | 84 (66.67%) | | 6(4.76%) | 120 (95.24%) | | 75 (59.52%) | 51 (40.48%) | | 93 (73.81%) | 33 (26.19%) | |
| | 35–44 | 30 (43.48%) | 39 (56.52%) | | 3(4.35%) | 66 (95.65%) | | 42 (60.87%) | 27 (39.13%) | | 30 (43.48%) | 39 (56.52%) | |
| | > 45 | 3(25%) | 9(75%) | | 0(0%) | 12(100%) | | 9(75%) | 3(25%) | | 9(75%) | 3(25%) | |
| Marital status | Single/ Divorced/ Widowed | 39 (33.33%) | 78 (66.67%) | 0.293 | 6(5.13%) | 111 (94.87%) | 0.490 | 60 (51.28%) | 57 (48.72%) | 0.152 | 69 (58.97%) | 48 (41.03%) | 0.274 |
| | Married/Live together | 45 (39.47%) | 69 (60.53%) | | 6(5.26%) | 108 (94.74%) | | 72 (63.16%) | 42 (36.84%) | | 75 (65.79%) | 39 (34.21%) | |
| Occupation | Household | 21 (33.33%) | 42 (66.67%) | 0.423 | 3(4.76%) | 60 (95.24%) | 0.590 | 45 (71.43%) | 18 (28.57%) | 0.244 | 42 (66.67%) | 21 (33.33%) | 0.965 |
| | Civil servant | 12 (22.22%) | 42 (77.78%) | | 6 (11.11%) | 48 (88.89%) | | 30 (55.56%) | 24 (44.44%) | | 33 (61.11%) | 21 (38.89%) | |
| | Shopkeeper | 9 (42.86%) | 12 (57.14%) | | 0(0%) | 21(100%) | | 6(28.57%) | 15 (71.43%) | | 12 (57.14%) | 9 (42.86%) | |
| | Others | 42 (45.16%) | 51 (54.84%) | | 3(3.23%) | 90 (96.77%) | | 51 (54.84%) | 42 (45.16%) | | 57 (61.29%) | 36 (38.71%) | |
| Education | No formal education | 0(0%) | 0(0%) | 0.704 | 0(0%) | 0(0%) | 0.801 | 0(0%) | 0(0%) | 0.897 | 0(0%) | 0(0%) | 0.022 |
| | Primary | 6(25%) | 18(75%) | | 0(0%) | 24(100%) | | 15 (62.50%) | 9 (37.50%) | | 6(25%) | 18(75%) | |
| | Secondary | 45(35%) | 78(65%) | | 12(5%) | 114(95%) | | 69(55%) | 54(45%) | | 75(60%) | 48(40%) | |
| | University | 33 (40.74%) | 51 (59.26%) | | 3(3.70%) | 81 (96.30%) | | 48 (59.26%) | 36 (40.74%) | | 63 (77.78%) | 21 (22.22%) | |
| Religion | Catholic | 48 (34.78%) | 90 (65.22%) | 0.117 | 6(4.35%) | 132 (95.65%) | 0.696 | 87 (63.04%) | 51 (36.96%) | 0.187 | 84 (60.87%) | 54 (39.13%) | 0.764 |
| | Protestant | 33(50%) | 33(50%) | | 3(4.55%) | 63 (95.45%) | | 27 (40.91%) | 39 (50.09%) | | 45 (68.18%) | 21 (31.82%) | |
| | Muslim and others | 3 (11.11%) | 24 (88.89%) | | 3 (11.11%) | 24 (88.89%) | | 18 (66.67%) | 9 (33.33%) | | 15 (55.56%) | 12 (44.44%) | |
| Type of contraceptive used | Implant | 27(30%) | 63(70%) | 0.626 | 9(10%) | 81(90%) | 0.300 | 42 (46.67%) | 48 (53.33%) | 0.326 | 42 (46.67%) | 48 (53.33%) | 0.071 |
| | IUD | 12 (44.44%) | 15 (55.56%) | | 0(0%) | 27(100%) | | 18 (66.67%) | 9 (33.33%) | | 21 (77.78%) | 6 (22.22%) | |
| | Injectable | 45 (39.47%) | 69 (60.53%) | | 3(2.63%) | 111 (97.37%) | | 72 (63.16%) | 42 (36.84%) | | 81 (71.05%) | 33 (28.95%) | |
| | Pills | 0(0%) | 0(0%) | | 0(0%) | 0(0%) | | 0(0%) | 0(0%) | | 0(0%) | 0(0%) | |
| **Total** | | **84 (36.36%)** | **147 (63.64%)** | | **12 (5.19%)** | **219 (94.81%)** | | **132 (57.14%)** | **99 (42.86%)** | | **144 (62.34%)** | **87 (37.66%)** | |

MetS = Metabolic Syndrome; IUD = Intra-uterine Device; % = Percentage; n: Size

## 3. Metabolic syndrome Prevalence according to the type of contraceptive used

The risk of MetS increases in women using injectable contraceptives (58.82%), which are the subjects with at least three components of metabolic syndrome. On the other hand, it is more manageable in those using intra-uterine devices (11.76%) and implants (29.42%) as shown in Fig 1.

**Table 3. Variables according to status of metabolic syndrome from NCEP-ATPIII criteria.**

| Variables | n (%) | In good Heath | | n (%) | Not in good Heath | | P-value |
|---|---|---|---|---|---|---|---|
| | | Mean ±SD | Median | | Mean ±SD | Median | |
| Glycemia (x 100 mg/dl) | 219 (94.80%) | 0.78 ±0.10 | 0.79 | 12 (5.20%) | 1.26 ±0.41 | 1.22 | 0.007 |
| TC (x 100 mg/dl) | 147 (63.63%) | 1.43 ±0.24 | 1.44 | 84 (36.37%) | 2.20 ±0.24 | 1.87 | 0.887 |
| HDL-c (x 100 mg/dl) | 114 (49.35%) | 0.58 ±0.07 | 0.56 | 117 (50.65%) | 0.45 ±0.05 | 0.47 | 0.038 |
| LDL (x 100 mg/dl) | 204 (88.31%) | 0.87 ±0.30 | 0.89 | 27 (11.69%) | 1.36 ±0.13 | 1.51 | 0.018 |
| TG (x 100 mg/dl) | 198 (85.71%) | 0.92 ±0.27 | 0.56 | 33 (14.29%) | 2.14 ±0.11 | 1.62 | 0.003 |
| SBP (mm Hg) | 219 (94.80%) | 110.10 ±10.32 | 111 | 12 (5.20%) | 141 ±7.35 | 141 | 0.489 |
| DBP (mm Hg) | 96 (41.56%) | 73.53 ±4.49 | 74 | 135 (58.44%) | 85.62 ±4.20 | 86 | 0.674 |
| Overweight (kg/m$^2$) | 141 (61.04%) | 24.16 ±5.19 | 24.22 | 90 (38.96%) | 27.21 ±1.57 | 26.73 | 0.039 |
| WC (cm) | 147 (63.63%) | 78.77 ±7.16 | 80 | 84 (36.37%) | 95.04 ±6.38 | 93 | 0.504 |

MetS = Metabolic Syndrome; SD = Standard deviation; TC = Total Cholesterol; HDL-c = High density lipoproteins cholesterol; LDL-c = Low Density Lipoproteins cholesterol; TG = Triglyceride; SBP = Systolic Blood Pressure

DBP = Diastolic Blood Pressure; WC = Waist Circumference; n = Size; % = Percentage

## 4. Overall prevalence of Metabolic syndrome of the study population

In the study population, as shown in Fig 2, exactly 22.08% women regardless of the type of contraceptive used were suffering from metabolic syndrome, the others (77.92%) did not suffer from it with at most two components of metabolic syndrome.

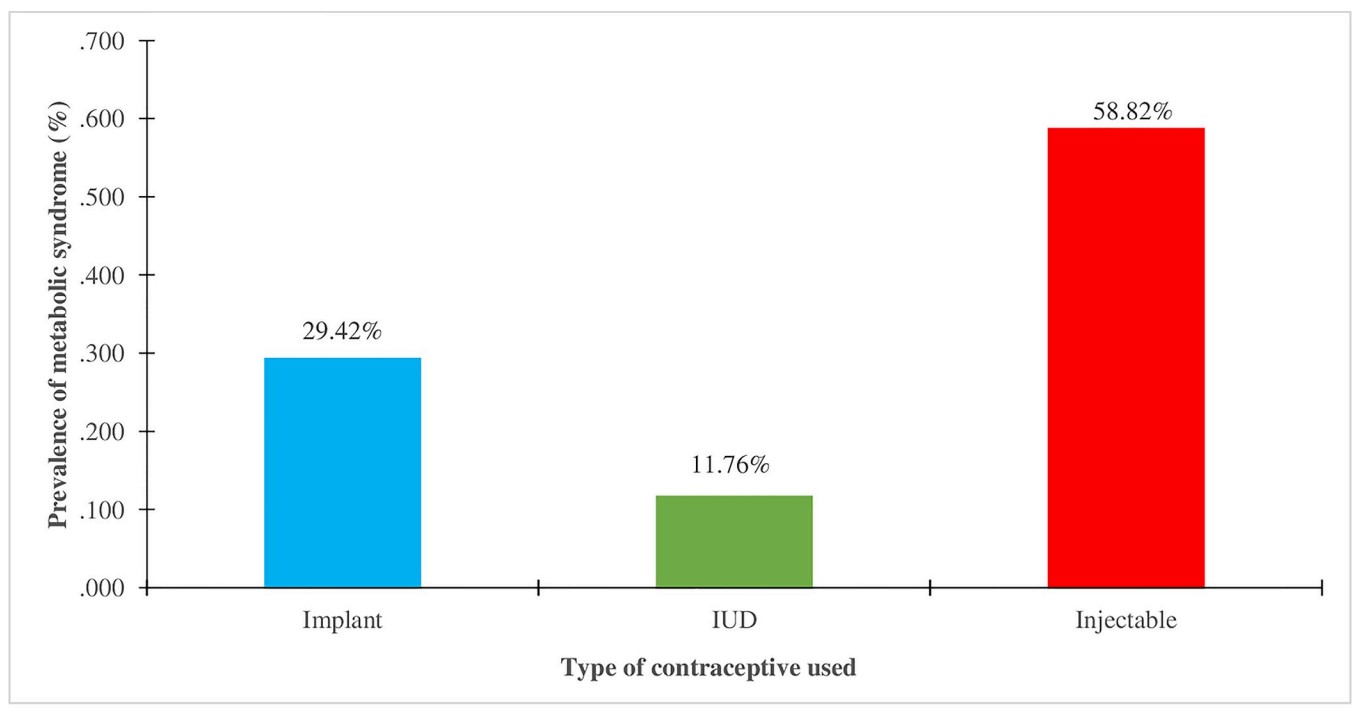

*IUD = Intra-uterine Device ; % = Percentage.*

**Fig 1. Prevalence of metabolic syndrome according to the type of contraceptive used.** IUD = Intra-uterine Device; % = Percentage.

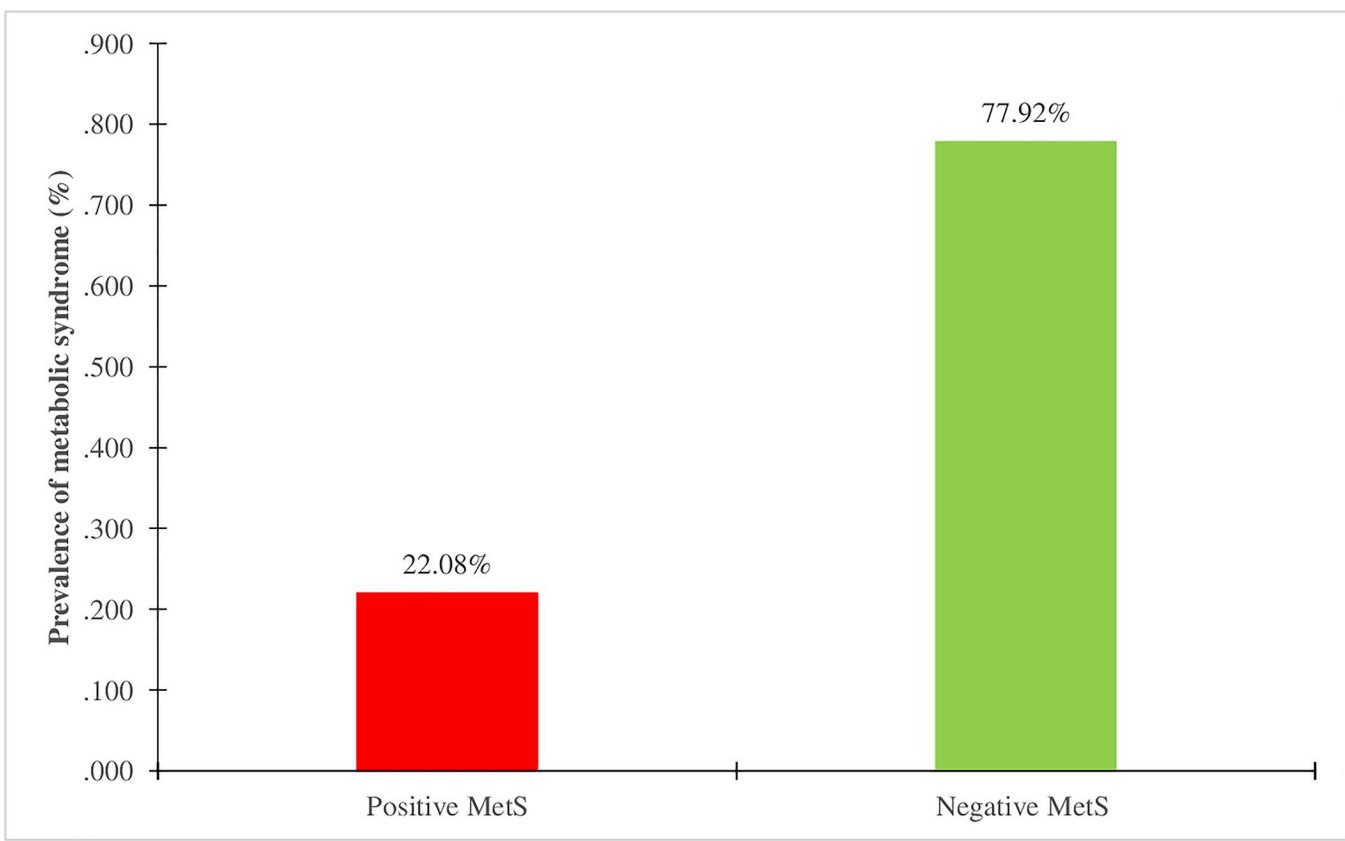

**Fig 2. Overall prevalence of metabolic syndrome.** MetS = Metabolic Syndrome; % = Percentage.

## Discussion

A total of 231 participants were included in the current study with women from 19 to 48 years in some district hospital in Douala-Cameroon with mean age of 32.30 ±7.60 years, many of them are using those patterns and trends of contraceptives. A total of 231 participants were using modern contraceptives methods, 28 were not using contraceptive methods and 12 (4.43%) were non-consenting to our study [15].

Table 2 shows that among these women, 50% of Protestants suffer from obesity and 66.67% of Muslims and others are hypertensive. We also note that university-educated women suffer more from hypertension (59.26%) and dyslipidemia (77.78%), while married women are more numerous suffering from dyslipidemia (65.79%). Similar results were obtained firstly, by Ajong *et al* who had found that the population was dominated by Christians (97.30%) and about 8 out of 10 women had at least a secondary education [16]. Secondly, by Njotang *et al*. showing that among women on contraceptives, 52.20% were unmarried and 86.8% of them had at least secondary education [17]. The very low proportion of other religions could be explained by the fact that in Cameroon, their opinions do not approve of contraceptive methods, and Christians are in the majority. The availability of housewives would explain their high rate of use compared to all women. The use of modern contraceptive methods among single and married women could be explained by the fact that married women consider it beneficial and vital to apply family planning rules by limiting and spacing births, and that single women make greater use of modern contraceptive methods to avoid unwanted pregnancies while preserving their image within society.

These results reveal a high rate (85.24%) of the use of modern contraceptive methods among these women, which is much higher than the prevalence rate of 58.90% reported by Njotang *et al* [17]. and 18.70% according to the 2020 annual family planning report [6]. The high rate of contraceptive use reported in our study can be explained by the fact that the survey was conducted in a family planning unit where most women are on contraceptives. In addition, the majority of contraceptive users preferred the injectable method, the one most used by the women in this study. This conclusion was also confirmed in a study conducted by Yangsi *et al*. in which the most commonly used contraceptive method was the injectable (72.10%) [18]. This may be due to the fact that the injectable method is more comfortable, less expensive and therefore easily accessible. This result is opposite to that reported by Tolefac et al. and Ajong et al. in Cameroon, where their study found that implants and the IUD were the contraceptive methods most used by 29.40% and 28.40% of participants respectively [15, 16]. Suggesting that their work in Yaoundé like the current study in Douala, took place in an urban area with the same lifestyle and characteristics. Similarly, women generally prefer this method, most probably because of its long period of action, compared with other methods which, like pills, require a strict daily intake.

The distribution of MetS determinants according to the NCEP ATP III giving to the different contraceptive methods [14] shows in Table 3 that 5.2% of the women had an elevated BP ($\geq$ 130 mmHg). Almost (36.37%) had a WC greater than 88 cm, and 50.65% of these women had an HDL-c level less than 40 mg/dl. A very small number of these women (5.20%) had elevated blood glucose and serum TG levels (14.29%). These results are similar to those of Cai *et al*. in their study of MetS in women in China who found that there were 52.60% of women with high WC according to the specific ethnic cut-off (80 cm for Chinese women), of which only 23.50% met the criteria for high WC. In addition, 22.90% of participants had low HDL-c levels, while 24.50% had high TG levels, 33.70% had hypertension and 6.6% had fasting blood glucose levels greater than 110 mg/dl [19]. Another study conducted in Nigeria by Sabir *et al*., in which the classification of MetS in women was based on NCEP ATP III guidelines, had found high WC, hypertension, high blood glucose, increased TG and low HDL-c; corresponding to 49.80%, 46.10%, 32.70%, 22.40% and 56.1% respectively [20]. There is a disparity between some of the data obtained from MetS determinants such as blood glucose and TG in our study and those obtained elsewhere. Nevertheless, data such as waist circumference, BP and HDL-c, are close to those of the other studies in spite of a larger sample size in the later study. However, in our study, the contraceptive method used would have little impact on the determinants of MetS.

It also appears in Fig 1 that, from the different types of modern contraceptives used by these women, the injectable method presents more danger, with a high MetS prevalence of 58.82%. This result suggests that, although this method is preferred by women, its secondary effects would be immediate and lead to a disruption of the body's metabolic balance with consequences. Our results on MetS also report in Fig 2 a prevalence of 22.80% among these women on contraceptives according to the NCEP-ATPIII criteria. This significative prevalence is similar to that obtained by Faijer *et al*. on a study of MetS in sub-Saharan Africa, where the overall prevalence varied according to the different criteria; according to the IDF: 18.00%, NCEP-ATP III: 17.10% and WHO: 11.10% [21, 22]. In another study conducted on MetS in women in China, prevalence according to NCEP-ATPIII was 16.90% [19]. Although these patterns of prevalence are similar, we however find that the prevalence of our study is higher than others. This would be due to the fact that in these studies the sample size is quite large and representative of the general population as well as the duration of the study which is quite long. Nevertheless, we found that contraceptive use had a minor impact and non-significant on the occurrence of MetS. This could suggest that the occurrence of MetS in these women is not

related to contraception alone but rather to their lifestyle habits such as diet and physical activity.

## Conclusion

This study was focused on women using modern contraception and the main objective was to investigate potential metabolic disorders associated with modern contraceptive use as risk factors for metabolic syndrome. It was found that, contraceptive use was certain in all the participants, and findings in this study reported, according to the NCEP-ATPIII a prevalence of 22.08% of metabolic syndrome among women using modern contraceptive methods in Douala, Republic of Cameroon. The high-risk groups were women using injectable method. Therefore, lipid profiles should be assessed in those women in order to manage them better.

## Supporting information

**S1 File. Raw data of the study.** HDL-c = High density lipoproteins cholesterol; LDL-c = Low Density Lipoproteins cholesterol; TG = Triglyceride; SBP = Systolic Blood Pressure; DBP = Diastolic Blood Pressure; BMI = Body Mass Index; HT = Herpertensive; WC = Waist Circumference.
(XLS)

## Acknowledgments

The authors would like to thank the participating patients and all the laboratory staff of Douala General Hospital who helped in the biochemical analysis of this data.

## Author Contributions

**Conceptualization:** Dandji Saah Marc Bertrand, Dangang Bossi Donald Séverin, Zambou Ngoufack François.

**Formal analysis:** Dandji Saah Marc Bertrand.

**Funding acquisition:** Dandji Saah Marc Bertrand, Zambou Ngoufack François.

**Investigation:** Dandji Saah Marc Bertrand, Dangang Bossi Donald Séverin, Tanguenan Floraise Lynda.

**Methodology:** Dandji Saah Marc Bertrand, Dangang Bossi Donald Séverin, Tanguenan Floraise Lynda, Zambou Ngoufack François.

**Project administration:** Dandji Saah Marc Bertrand, Zambou Ngoufack François.

**Software:** Dandji Saah Marc Bertrand, Dangang Bossi Donald Séverin, Tanguenan Floraise Lynda.

**Supervision:** Dandji Saah Marc Bertrand, Zambou Ngoufack François.

**Validation:** Dandji Saah Marc Bertrand, Dangang Bossi Donald Séverin, Zambou Ngoufack François.

**Visualization:** Dandji Saah Marc Bertrand, Dangang Bossi Donald Séverin, Tanguenan Floraise Lynda, Zambou Ngoufack François.

**Writing – original draft:** Dandji Saah Marc Bertrand, Dangang Bossi Donald Séverin, Tanguenan Floraise Lynda, Zambou Ngoufack François.

**Writing – review & editing:** Dandji Saah Marc Bertrand, Dangang Bossi Donald Séverin, Zambou Ngoufack François.

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
