## [Decision Letter · Decision Letter 0]

20 Mar 2024

PONE-D-23-31347Metabolic syndrome and related factors in Cameroonian women under contraceptive usePLOS ONE

Dear Dr. Marc Bertrand,

Thank you for submitting your manuscript to PLOS ONE. After careful consideration, we feel that it has merit but does not fully meet PLOS ONE’s publication criteria as it currently stands. Therefore, we invite you to submit a revised version of the manuscript that addresses the points raised during the review process.

We look forward to receiving your revised manuscript.

Kind regards,

Ibrahim Sebutu Bello, MBBS, MPH, MD, FMCGP

Academic Editor

PLOS ONE

Reviewers' comments:

Reviewer's Responses to Questions

**Comments to the Author**

1. Is the manuscript technically sound, and do the data support the conclusions?

Reviewer #1: No

Reviewer #2: Partly

2. Has the statistical analysis been performed appropriately and rigorously? 

Reviewer #1: No

Reviewer #2: No

3. Have the authors made all data underlying the findings in their manuscript fully available?

Reviewer #1: Yes

Reviewer #2: Yes

4. Is the manuscript presented in an intelligible fashion and written in standard English?

Reviewer #1: No

Reviewer #2: No

5. Review Comments to the Author

Reviewer #1: Title: Metabolic syndrome and related factors in Cameroonian women under contraceptive use

General comments.

The manuscript sought to address an important topical issue. However, there are many critical flaws and inconsistencies with regard to the objective, design, and results of the study. A comprehensive reconstruction of the article will be required. There are also lots of errors in grammar and sentence construction. A professional proofreading will be necessary.

Authors should also ensure the presentation of the manuscript and the references are in keeping with the journal requirements.

Specific issues

Abstract

Page 2, lines 12-13: “Contraceptives can cause a myriad of secondary effects such as overweight, increase of blood pressure and triglycerides, and carbohydrate intolerance.” This is not true for all contraceptives. It is true for some types. Carbohydrate intolerance is the inability to digest certain carbohydrates due to a lack of certain enzymes. Glucose intolerance will be a more appropriate term here.

Lines 46 – 50: The use of 2012 data is very old for such an important statistic that is usually regularly updated. The same applies to reference 10 in lines 70-72, and some others.

Lines 77-79: A cross-sectional study design cannot determine ‘effect’ (i.e. causality). It can at best determine associations.

Materials and methods

Study population: The description of the target and study population is unclear and confusing. If the target population were women on contraceptives, as stated by authors, the study population and sample can therefore not include women not on contraceptives.

Lines 92-93: “Participants were 93 recruited by the principal investigator in the various health facilities…” The details of the sampling technique should be stated.

Lines 94-96: “We have contact 271 women, of whom 231 were on modern contraceptives, 28 were not using contraceptives, and 12 did not consent to the study”. This statement is confusing. For example, if 12 women did not consent, how come authors kept repeating in different aspects of the manuscript that 271 were INCLUDED in the study? Furthermore, based on what the authors stated to be the target population, there cannot be inclusion of 28 who did not meet that criteria.

How many respondents were in the study i.e. to whom the questionnaire was administered and had their anthropometrics data, lifestyle factors, and biochemical parameters collected?

Pg 7, Lines 132 – 135: “To measure the WC, the tape was positioned at the natural waist or at the lower curvature located between the last rib and the iliac crest”. Please clarify the meaning of this statement, especially ‘natural waist’ and ‘lower curvature’.

Lines 141-142: “Elevated BP as a component of MetS was 142 defined as ≥ 130/80 mmHg”. This contradicts earlier stated criteria viz lines 19-20, and 64-65.

Lines 152-155: “Each blood sample was labelled with the participants’ number to avoid errors of recording. Blood glucose values were defined as normal ˂ 1.1 g/l; limit 1 – 1.25 g/l and diabetic ˃ 1.25 g/l. Raised TG was defined as TG level 0.4 – 1.4 g/l irrespective of gender and hypertriglyceridemia at ˃ 1.4 g/l. Whereas, low HDL-c level was defined as HDL-c ˃ 0.5 g/l in women and hypocholesterolaemia at ˂ 0.5 g/l”. The statement is confusing. For example, what is the difference between ‘raised TG” and ‘hypertriglyceridemia’, what is ‘hypocholesterolaemia’ etc.

Statistical analysis: Chi-square was used to analyse which variables? And Mann-Whitney for which type of variables?

Results

Line 173: It is incorrect to say 271 were surveyed or that 271 were included, for reasons earlier highlighted. It is also not possible to assess the prevalence of contraceptive use because, according to the authors, the target population (hence inclusion criteria) was to consist only of those on contraceptives

Lines 176-177: “The background characteristics of this study population are shown in Table 1”. Table 1 does not however contain the prevalence of each sub-category of the background characteristics.

Page 10, line 181: What is the definition of ‘healthy’ and ‘unhealthy’ subjects

Line 181-183: “. Out of two hundred and thirty-one women, 36.37% (95.04 ±6.38) were obese; 58.44% (85.62 ±4.20) were hypertensive (DBP) and 50.65% (0.45 ±0.05) had dyslipidemia due to HDL-c imbalance…”. What are the figures in brackets? Also, the prevalence of hypertension and dyslipidemia stated here are different from the ones in Table 1.

Lines 185-187: The meaning of the following statement is unclear: “The means (±SD) of the components of MetS including other related variables like overweight, systolic blood pressure, total cholesterol, LDL-c, and triglycerides values were demonstrated among the study population”

Table 2: it is difficult to appreciate what the purpose of this table is and the interpretation of the information contained therein. There are several issues with it but here are some examples: what is the meaning of ‘healthy’ or ‘unhealthy’? The n(%) for each of the variables is referring to what? What is ‘glycemia’? The p-values are tests of significance for which statistical test? What is the contribution of the table to the objective of the article? Etc.

Lines 191-193: “The risk of MetS increases in women using injectable contraceptives (58.82%), which are the subjects with at least three components of metabolic syndrome. It is otherwise controllable in those using the Intra-uterine device (11.76%) and implant (29.42%)…”. Like many such statements in different aspects of the manuscript, some of which have been highlighted earlier, this statement is difficult to understand.

What is being shown in Figure 1 is also unclear!

It is pertinent to note that there is nothing in the results presented that showed an analysis of any association between the use of contraceptives and MetS, in line with the objective of the article. In any case, that will not be possible given that the target population was meant to include only those on contraceptives.

Discussion

The discussion is difficult to follow and has some specific issues, which are not limited to the following:

1. There are lots of discrepancies between the facts being discussed and the results earlier presented, for example, in lines 206-208: “Table 1 shows that most of these women were Christians (88.31%), 50.65% of them were 207 single, only 27.27% were housewives, and more than half had acquired at least a secondary 208 education (53.25%)” There is no such information on table 1.

2. Another example in line 251: ‘However, in our study, the contraceptive method used had little impact on the determinants of MetS.” The is no data reflecting this in the results and, in any case, the study design cannot make such a determination.

3. The discussion around the ‘prevalence’ of the use of contraceptives becomes superfluous in view of the reasons earlier mentioned i.e. the design of the study cannot determine that.

Conclusion: Given the issues raised regarding the methods and results, the study can not make the conclusions stated. Furthermore, there is no data presented on the analysis of dietary profile (line 278)

Reviewer #2: Line 17: This number seems to be different from what you have in the results section. 271 vs. 231.

Line 22: Why are those who were not contraceptives and those who did not consent being reported as part of the total sample size. The study design was a cross- sectional one so how does these different categories fit in?

Line 27-28: Metabolic syndrome consist of a myriad of conditions and the confounders are quite a lot. How were the author able to eliminate those confounders. Women on contraceptives can be exposed to other conditions that can predispose them to metabolic syndrome? So, what were the authors trying to achieve by measuring metabolic syndrome in such women? Then again, the best approach will have been to use two arms of women i.e., women on contraceptives and those not on it. This should have been age-matched and sample matched. Then we could have made some inferences from the results. As it stands now, there are a lot pf confounding factors that can lead to a bias in the results.

Line 29: The focus of the work was to look metabolic syndrome in women on contraceptive. Now in the conclusion the authors seems to be silent on this. Is there a reason?

Line 32-33: How did the authors arrive at this conclusion? No part of the work looked at dietary pattern.

Line 77-78: How was this done? How was the effect of contraceptive on metabolic syndrome done in this study? And how were the other confounders controlled for?

Line 81: In the abstract, it was stated that 231 women were on contraceptive. So where from this (271)? Why was a cross-sectional study used?

Line 95: So what happened to the 12 that did not consent? were they added or excluded from the study?

Line 116: Did the patients fast before the sample was taken?

Line 119-120: By what method was the glucose determined?

Line 149-150: This has already been described under line 118-119. Please delete

Line 152-155: Glucose measurements are usually done in mmol/l or mg/dl. Please revise. Same for TG and HDL-c

Line 159: Why was the Mann-Whitney test used?

Line 173: This sample size is misleading. Please revise.

Line 173-177: The descriptions given here cannot be found in the table so why is reference made to the table 1?

Line 178-179: Table 1 is very confusing because of the stratification that was done. Why were the stratifications like obese, diabetic etc done? This makes it difficult to understand the data presented. The authors could have just presented the general demography without any stratification for readers to appreciate the dynamics of the studied population. As it stands now it’s very difficult to make any meaning from the table. Please revise

Line 181: What was used to classify the subjects as either healthy or unhealthy?

Line 182: How come the authors are now using 231 in their data analysis? What happened to those not on contraceptives or did not consent?

Line 184: Are these women not those classified as healthy?

Line 188: What was the criteria for classifying the women as health and unhealthy in table 2? The main focus of the work is on contraceptive use and Metabolic syndrome. So, what is the essence of this stratification? Why was mean and median measured in table 2?

The unit of measurement for the glucose and the lipid profile are not standard. Please use the WHO approved units of measurement. Ref Table 2.

The discussion section needs to be looked at again. There seem to be no coherence in syncing the results with the discussion.

Line 201: why are the sample size still 271?

Line 212-218. Provide supporting references for the statements made.

Line 228-235: Statements not so clear. Please revise.

Line 238: Why is 36.37% being described as 'many'?

Line 253-258: The reasons given for why the injectable method presents with a higher metabolic syndrome is not backed by literature in the discussion section. What is the pathophysiology of the effect of injectables on metabolic syndrome?

Line 278: Was there any analysis of dietary habits in relation to metabolic syndrome? The conclusion needs to be written again with focus on the major findings of the study.

6. PLOS authors have the option to publish the peer review history of their article (what does this mean?). If published, this will include your full peer review and any attached files.

Reviewer #1: No

Reviewer #2: No

---

## [Author Response · Author response to Decision Letter 0]

6 Jun 2024

I've already respond on the Response to Reviewers sheet

---

## [Decision Letter · Decision Letter 1]

6 Aug 2024

Metabolic syndrome and related factors in Cameroonian women under contraceptive use

PONE-D-23-31347R1

Dear Dr. Marc Bertrand,

We’re pleased to inform you that your manuscript has been judged scientifically suitable for publication and will be formally accepted for publication once it meets all outstanding technical requirements.

Kind regards,

Ibrahim Sebutu Bello, MBBS, MPH, MD, FMCGP

Academic Editor

PLOS ONE

Reviewers' comments:

Reviewer's Responses to Questions

**Comments to the Author**

1. If the authors have adequately addressed your comments raised in a previous round of review and you feel that this manuscript is now acceptable for publication, you may indicate that here to bypass the “Comments to the Author” section, enter your conflict of interest statement in the “Confidential to Editor” section, and submit your "Accept" recommendation.

Reviewer #1: (No Response)

Reviewer #2: All comments have been addressed

2. Is the manuscript technically sound, and do the data support the conclusions?

Reviewer #1: Partly

Reviewer #2: Yes

3. Has the statistical analysis been performed appropriately and rigorously? 

Reviewer #1: I Don't Know

Reviewer #2: Yes

4. Have the authors made all data underlying the findings in their manuscript fully available?

Reviewer #1: (No Response)

Reviewer #2: Yes

5. Is the manuscript presented in an intelligible fashion and written in standard English?

Reviewer #1: (No Response)

Reviewer #2: Yes

6. Review Comments to the Author

Reviewer #1: I wish to commend the authors for the time and effort taken to incorporate many of the corrections and suggestions.

However, it is important to reiterate that, based on the fact that total sample size, on whom the data collection tool was applied, equals 231, and all were contraceptive users, there is therefore no basis for calculating the prevalence of contraceptive use.

Reviewer #2: (No Response)

7. PLOS authors have the option to publish the peer review history of their article (what does this mean?). If published, this will include your full peer review and any attached files.

Reviewer #1: No

Reviewer #2: **Yes: **Christian Obirikorang

---

## [Editor Report · Acceptance letter]

15 Nov 2024

PONE-D-23-31347R1 

PLOS ONE

Dear Dr. Marc Bertrand, 

I'm pleased to inform you that your manuscript has been deemed suitable for publication in PLOS ONE. Congratulations! Your manuscript is now being handed over to our production team.

Kind regards, 

on behalf of

Dr. Ibrahim Sebutu Bello 

Academic Editor

PLOS ONE